# Effect of 1,25-Dihydroxyvitamin D3-Glycosides on the Farrowing Process and Piglet Vitality in a Free Farrowing System

**DOI:** 10.3390/ani12050611

**Published:** 2022-02-28

**Authors:** Laura Jahn, Gertraud Schuepbach-Regula, Heiko Nathues, Alexander Grahofer

**Affiliations:** 1Clinic for Swine, Department for Clinical Veterinary Medicine, Vetsuisse Faculty, University of Bern, 3012 Bern, Switzerland; laura.jahn@vetsuisse.unibe.ch (L.J.); heiko.nathues@vetsuisse.unibe.ch (H.N.); 2Department of Clinical Research and Veterinary Public Health, Vetsuisse Faculty, Veterinary Public Health Institute, University of Bern, 3012 Bern, Switzerland; gertraud.schuepbach@vetsuisse.unibe.ch

**Keywords:** sow, parturition, umbilical cord, placenta, litter size, litter weight

## Abstract

**Simple Summary:**

Vitamin D improves the reproductive performance in animals. This study aimed to examine the effects of 1,25-dihydroxyvitamin D3-glycosides (1,25-vitD) on the farrowing process in sows and the vitality of their piglets. Therefore, 100 sows were divided into two groups at insemination (‘1,25-vitD’ and ‘negative control’). The farrowing process and the piglet vitality was evaluated. The number of born piglets in sows of ‘1,25-vitD’ was higher and the farrowing duration was shorter than in the negative control group without showing significance in the univariable analysis. In a further statistical model including the variables ‘farrowing duration’, ‘total born piglets’ and ‘1,25-vitD’ differences became evident. We found that 1,25-vitD was associated with a reduced farrowing duration (*p* = 0.055). Moreover, significantly more mummies (*p* < 0.01) and short ruptured umbilical cords (*p* < 0.05) were observed in the 1,25-vitD group. This study showed an effect of 1,25-vitD on the farrowing process.

**Abstract:**

Vitamin D improves the reproductive efficiency in animals. This study aimed to examine the effects of 1,25-dihydroxyvitamin D3-gylcosides (1,25-vitD) on the farrowing process in sows and the vitality of their piglets. In total, 100 sows were allocated into two groups at insemination (‘1,25-vitD’ and ‘negative control’). The 1,25-vitD group received 260–300 mg/sow/day 1,25-vitD in their feed during the gestation period. Backfat thickness, fecal score, and the farrowing process was evaluated. The piglets were categorized into live born or stillborn, and vitality was evaluated by assessing the umbilical cord and the meconium score. The number of total-born piglets in sows of ‘1,25-vitD’ was higher and the farrowing duration was shorter than in the negative control group without showing significance in the univariable analysis. In a linear multiple regression model including the variables ‘farrowing duration’, ‘total born piglets’ and ‘1,25-vitD’ differences became evident. We found that 1,25-vitD was associated with a reduced farrowing duration (*p* = 0.055). Moreover, significantly more mummies (*p* < 0.01) and short ruptured umbilical cords (*p* < 0.05) were observed in the 1,25-vitD group. This study showed an effect of 1,25-vitD on the farrowing process. However, more research is needed to better describe the mechanism of 1,25-vitD in detail.

## 1. Introduction

The role of vitamin D to improve reproductive efficiency has become known in animals and humans in recent years [1,2,3,4,5]. It has been described that mating success, litter size, and neonatal growth were reduced in rats with vitamin D deficiency [2,6]. Furthermore, low vitamin D concentrations were associated with various reproductive disorders, such as polycystic ovary syndrome and infertility [7]. In addition, a better reproductive performance (outcome of in vitro fertilization) has been detected in women with adequate vitamin D concentrations [7]. In pigs, it is described that vitamin D has an effect on fertility during gestation [8]. On the other hand, it is known that an overdose of vitamin D is fetotoxic in mammals [9], and can lead to abortion at different stages of pregnancy in sows [10]. Overall, it is evident that vitamin D influences the reproductive tract and reproductive performance in mammals [1,2,3,8,11].

Improvement of sows’ health during gestation and parturition is an important aspect of animal welfare and farm profitability and also has a tremendous influence on piglet vitality [12,13,14,15,16,17].

Due to an increase in litter size in the last decades, the vitality of piglets reduced, because of prolonged farrowing durations [16,18,19,20,21,22] and due to the effects of intrauterine growth restrictions [23,24,25]. Therefore, it is necessary to find measures to improve the farrowing process in hyperprolific sows and thereby increase piglets’ health and welfare. Various feed additives and supplements have already been tested to determine their influence on the mortality rate of piglets [26]. In a recent study, it was observed that the additional administration of 1,25(OH)_2_D3-gly (1,25-vitD) has a positive effect on the number of piglets born alive [27]. The authors hypothesized that 1,25-vitD improves the farrowing process in sows [28,29], which can positively influence the vitality of newborn piglets [30,31]. On the other hand, the number of stillborn piglets decreased with an increased dose of vitamin D and the bioavailability of 25-hydroxyvitamin D was higher compared to vitamin D [29,32]. As a natural source of the active vitamin D form calcitriol (1,25(OH)_2_vitamin D) Solanum glaucophyllum can be used [33], which is the 1,25-vitD source of Panbonis^®^. Until today, different dosages of vitamin D and calcidiol (25-hydroxyvitamin D) have been tested in gestating sows and growing pigs, leading to increased serum calcidiol. This did not affect maternal performance and litter characteristics, but average daily gain was improved in weaned piglets from sows fed calcidiol [34,35].

For piglets, the farrowing process itself is a risk of intrapartum death or being born with low vitality and, thus, having lower chances of survival [12,14,30,36,37,38]. Hypoxia at birth is a major stress factor that can also be associated with stillbirth. This has been associated with meconium staining [14,39] and short ruptured umbilical cords [40,41]. Further knowledge of 1,25-vitD influence on these important parturition and piglet vitality parameters is needed. Therefore, the aim of the study was to investigate the effect of 1,25-vitD on the farrowing process and piglet vitality in a free farrowing system.

## 2. Materials and Methods

### 2.1. Ethical Issues

The study protocol was approved by the responsible Cantonal Veterinary Office of Solothurn, Switzerland (license no. SO 02/2020; No. 33057).

### 2.2. Animals and Housing Conditions

In this experiment, 100 sows (Swiss Large White × Landrace) from five consecutive farrowing batches of a Swiss sow pool system working in a three-week batch farrowing were used. The sows were artificially inseminated by the farmer, and subsequently accommodated under similar housing conditions during the entire study period. Within the first week after insemination, the sows were transported to the gestation unit. Approximately at day 110 of gestation, the sows were moved to the farrowing unit, which was equipped with free farrowing pens that had partially slatted floors and a size of 2.15 × 2.60 m (5.59 m^2^). In the farrowing unit, sows received liquid feed twice a day and had free access to water via a drinking bowl with a water flow rate of more than 3.5 L per minute. In addition, the sows received approximately 1 kg straw per day as rooting and nest building material. During parturition, artificial light was turned on to enable farrowing observation and the room temperature was approximately 20 °C.

### 2.3. Parameters Evaluated

#### 2.3.1. Sows and Farrowing Traits

After entering the farrowing unit, the fecal score, body condition score, and sows’ backfat thickness were assessed one to five days before the expected farrowing date. The body condition score (BCS) was always determined by the same investigator with a one to five scoring system according to Muirhead [42]. Backfat thickness (BF) was measured at P2 position at the level of the last rib 45 mm, 65 mm, and 80 mm from the midline on both sides using ultrasonography (MyLab™OneVet, Esoate, Genoa, Italy). The fecal score (FS) was evaluated with a previously published score ranging from zero to five [43].

Retrospectively, the total duration of farrowing (time from the first piglet to the last placenta expelled), the piglet expulsion duration (time interval between the first and the last piglet expelled), the placenta expulsion duration (time interval between the expulsion of the first and the last placenta), and the piglet interval (time interval between expulsion of two successive live or stillborn piglets) was assessed. Piglet expulsion duration was grouped into greater than 300 min (PIGLONG) and less than or equal to 300 min (PIGSHORT) [44]. Additionally, the following parameters were determined: First piglet–first placenta (time interval between the expulsion of the first placenta and the first piglet), last piglet–last placenta (time interval between the expulsion of the last placenta and the last piglet, and last piglet–first placenta (time interval between the expulsion of the first placenta and the last piglet). The interval of first placenta expulsion and last piglet expulsion showed a negative value if the first placenta was expelled before the last piglet, value zero when they were simultaneously expelled, and a positive value if the first placenta was expelled after the last piglet. Further data, such as parity of sows or sows’ age, were collected from the electronic sow planner used in this sow pool system.

#### 2.3.2. Piglets Traits

The piglets were characterized as live born, pre-partum death (stillborn type 1), or intra-partum death (stillborn type 2). Total born piglets were determined from the sum of live born piglets, stillborn piglets type 1, and stillborn piglets type 2. In addition, a modified meconium score arranged in three different categories of meconium staining on the skin modified by Mota-Rojas [39] was employed. Piglets with meconium score one (MEC1) had no meconium staining on their skin at all. In piglets with meconium score two (MEC2) a moderate meconium staining was observed (<2/3 of the piglet’s body surface covered), and piglets with meconium score three (MEC3) were severely meconium stained (≥2/3 of the piglet’s body surface covered). Furthermore, the integrity (intact or ruptured umbilical cord after expulsion of the piglet) and the length (measured from the umbilicus to the rupture on the maternal side grouped into ≤15 cm or >15 cm) of the umbilical cord was evaluated.

After parturition, total litter weight (live born piglets, kg) and total placenta weight (kg) were determined by using a platform scale (Professional 3700, Soehnle Industrial Solutions GmbH, Backnang, Germany).

### 2.4. Experimental Design

In this experiment, a randomized parallel study design was applied in a Swiss sow pool system to investigate the effect of 1,25-vitD on the reproductive performance and the farrowing process of 100 sows.

In each of five insemination batches, sows were randomly assigned using a coin toss to the negative control or treatment group with 50 individuals per group. The first examination of sows took place in the farrowing unit always by the same investigator. Sows with a poor general health condition or a severe lameness were excluded from the study population after entering the farrowing unit. The negative control group (C) received commercial feed (31 MJ/d/sow to 40 MJ/d/sow, 1000 IU/kg Vitamin D3). The other group (1,25-vitD) received the same commercial feed with a defined amount of feed additive that contained a standardized level of 1,25(OH)_2_D3-gly from dried and ground Solanum glaucophyllum leaves (Herbonis Panbonis^®^, August, Switzerland) with of 26 g/sow/day (260 mg/sow/day) from day one to day 84 of gestation and 30 g/sow/day (300 mg/sow/day) from day 85 of gestation until farrowing. These dosages correspond to the ones in the registration of the product (CH-BIO-004 IMO). The ingredients were waxy-leaf nightshade meal, wheat middlings, and pregelatinized wheat starch as carrier. In the service and the farrowing unit group 1,25-vitD received the feed additive manually once a day. In the gestation unit, 1,25-vitD was added with a dosimeter installed in the electronic sow feeding system.

In the farrowing unit, sows were monitored for signs of impending parturition (nest-building behavior) by three previously trained investigators in shift work approximately two days before the expected farrowing date (day 115 of gestation). From the onset of farrowing (expulsion of first piglet) continuous supervision at least every 10 min was conducted 24 h per day if necessary (morning-, afternoon-, and nightshift). All data described above were collected digitally with an automatic time tracking system. 

In case of dystocia (piglet-to-piglet interval ≥60 min) obstetrical intervention (OI) was conducted. If a fetus was present in the vaginal or cervical canal, then the fetus was extracted manually and 20 I.U. of oxytocin were injected intramuscularly. If no fetus was present in the birth canal, then only oxytocin was administered.

### 2.5. Statistical Analyses

Data were collected digitally with a tablet using a database program (Microsoft Access 2016, Redmond, WA, USA) and then transferred into a spreadsheet program (Microsoft Excel 2016, Redmond, WA, USA). Statistical processing of all data was conducted using the software NCSS 12 (NCSS, LLC. Kaysville, UT, USA) and Stata 16 (StataCorp LLC, College Station, TX, USA).

The entire study population was examined descriptively as one study group and thereafter according to the treatments (1,25-vitD vs. negative control). All continuous data were tested for normality using the Shapiro–Wilk normality test. If there was no normal distribution, data were logarithm transformed and tested again for normality. For univariable analysis, the Equal Variance *t*-test was applied in case of normal distribution and equal variance. For normally distributed data with no equal variance, the Aspin–Welsh unequal-Variance *t*-test was used. Non-normally distributed data were analyzed using the Wilcoxon Rank-Sum test. For describing the umbilical cord integrity and length as well as the meconium score on herd level, they were converted from the individual values of the piglets into percent of the sow’s litter. The litter weight was divided by live born piglets to compare the average individual piglet weight between the groups.

For further use, the mean value of backfat measurement was determined for each sow considering the three different measurements of each pig. The first placenta expulsion to last piglet expulsion was grouped into BEFORE (negative values) and AFTER (zero and positive values) to test the influence on farrowing parameters (piglet expulsion duration and placenta expulsion duration) and a possible relation to the treatment groups.

The umbilical cord integrity and length as well as the meconium score were analyzed with a mixed-effect logistic regression, to show differences between the two treatment groups with the sow as random effect. Therefore, the meconium scores were grouped into ‘no meconium’ including MEC1 and ‘meconium-stained piglets’ (MEC2 and MEC3).

A linear multiple regression model was used to test total duration of farrowing in relation to feed additive (1,25-vitD/C) and total born piglets. To fulfill model assumptions of normality of residuals, the logarithm transformed farrowing duration was used. The regression coefficient was reversed with the inverse function 10^×^ for better interpretation.

All tests were determined to be statistically significant with a confidence interval of 95%, and *p*-values less than 0.05.

## 3. Results

In total, five farrowing batches with 89 Swiss Large White x Swiss Landrace sows and their piglets (*n* = 1.385) were included. Group 1,25-vitD contained of 48 sows with 767 piglets and in group C 41 sows with 618 piglets were included. Parity ranged from one to ten with a median of four. Group 1,25-vitD had a median parity of four (range: two to ten) and group C had a median parity of 4 (range: one to ten). The median age was 2.1 years (range: 1.1 to 4.6), where in group 1,25-vitD the median age was 2.1 years (range: 1.2 to 4.4 years), and in group C the median age was 2.3 years (range: 1.1 to 4.6 years). The median body weight was 260 kg (min: 204; max: 338 kg). Sows in group 1,25-vitD had a median body weight of 261 kg (min: 204; max: 315 kg), and sows in group C had a median body weight of 258 kg (min: 209; max: 338 kg). Between both treatment groups no significant differences in parity, age, and body weight were observed.

### 3.1. Sow and Farrowing Traits

Back fat thickness was normally distributed with an overall mean of 15.1 ± 4.1 mm. The mean backfat thickness was 15.4 ± 4.0 mm in group 1,25-vitD and 14.8 ± 4.3 mm in group C. The fecal score ranged from zero to four and the distribution of these parameter in the two treatment groups is presented in Table 1. There were no significant differences in backfat thickness or fecal score between the two treatment groups.

The mean total duration of farrowing in group 1,25-vitD was 494 ± 220 min, whereas group C had a total duration of farrowing of 586 ± 304 min. Piglet interval differed from 17.9 ± 7.0 min (1,25-vitD) to 19.5 ± 10.4 min (C). Comparison of farrowing times is presented in Figure 1.

In univariable analyses, the differences in the farrowing process between the two groups were not statistically significant. All single parameters are shown in Table 2.

### 3.2. Piglets Traits

An overview of the piglets’ traits is presented in Table 3. No significant differences between the total, live born piglets, and stillborn type 2 and group 1,25-vitD and group C could be detected. However, the group 1,25-vitD had significantly (*p* ≤ 0.01) more stillborn piglets type 1 compared to the negative control group.

Furthermore, a significantly higher percentage of short umbilical cords (≤15 cm) could be detected in the 1,25-vitD piglets compared to the control piglets, whereas the umbilical cord integrity showed no significant difference (*p* = 0.17). In contrast, less meconium staining was recorded in the 1,25-vitD group compared to the control group (*p* < 0.01).

### 3.3. Further Statistical Models of the Two Treatment Groups (1,25-vitD vs. C) on the Sow and Piglets Traits

The multiple regression model showed that farrowing with more total born piglets had a higher probability to last longer (*p* < 0.01) and the feed additive 1,25-vitD tended to decrease the farrowing duration (*p* = 0.055). Detailed information of the multiple regression model is presented in Table 4.

In the mixed-effect logistic regression with the sow as random effect for umbilical cord length, umbilical cord integrity, and grouped meconium score, only umbilical cord length stayed significantly different between group 1,25-vitD and the control group (*p* < 0.05). Umbilical cord integrity (*p* = 0.38) and meconium score (*p* = 0.19) showed no significant difference in this model. Detailed information is shown in Table 5.

## 4. Discussion

This is the first study evaluating the effect of 1,25-vitD supplementation on the farrowing process and piglet vitality at farrowing time in a free farrowing system. The randomized parallel design was chosen because it enables a reliable investigation of the effect of a feed additive on parturition and piglet vitality. To reduce the performance and attrition bias, three investigators conducted the sampling after training, and used a detailed evaluation form. Even though 89 sows were included in this study, the sample size per group was relatively small and, therefore, additional stratification, e.g., for parity, was not possible. Another limitation of this study was that only one specific population—from one Swiss sow pool system—was examined. Since only one sow pool system was included in the study, the housing and management conditions for all sows were identical, thereby allowing a valid comparison of the treatment groups, i.e., the effect of the treatment. Even though, free farrowing sows were used in this study, the reproductive performance can be extrapolated to larger populations in other countries, because the housing situation of the sows during gestation is comparable to other countries. Further to the study objectives, new data of back fat thickness and fecal score on the parturition process and piglet vitality in free farrowing sows were obtained.

In comparison with former studies [4,27], no significant difference in the litter size between the two treatment groups were detected in our study. It might be that the normal feeding ratio containing 1000 IU vitamin D/kg feed was sufficient to prevent vitamin D deficiency in the study population. Hence, it is known from the literature, that a vitamin deficiency in rats significantly reduces the litter size [2,6]. In addition, the small sample size per group and the large litter size (15 live born piglets) in the studied sow population in comparison with the Swiss herd book 2020 (13 live born piglets) [45] might also have diminished the effect of 1,25-vitD on the litter size. Vitamin D plays an important role in calcium–phosphate homeostasis [5], and calcium ions are essential for uterine muscle contractions [46], influencing the piglet and placental expulsion duration [47]. A current study showed in a cross-over design that 1,25-vitD significantly shortened the farrowing process from 370 to 256 min [28]. Another study only indicated that the piglet expulsion was shorter in the vitamin D group compared to the control group (280 min, 323 min) [29]. However, in contrast with literature, in the multiple regression model including the litter size as a dependent variable, only a trend (*p* = 0.055) in decreasing the total farrowing duration by using the feed additive 1,25-vitD could be detected. Although this effect is not statistically significant, the observed difference of 92 min can have clinical relevance for improving the parturition process in sows, particularly evaluating only this small study population [48]. Furthermore, in both groups, the piglet expulsion duration was approximately 250 min, which is almost an hour below the cut-off value used for declaring a ‘prolonged farrowing in sows’ [44], which indicates an optimal farrowing process on the farm. However, a longer farrowing duration leads to an increase in stillborn piglets [20] and, therefore, negatively influences the reproductive performance in a sow herd. Depending on the current knowledge, 1,25-vitD might influence the farrowing duration, but warrants further investigation. For further studies, the vitamin D receptor, which is expressed in the endometrium of the sow and the chorion of the embryo during gestation [8], might add further knowledge of the vitamin D effect on fertility in sows.

Due to an increased litter size and prolonged farrowing, the piglet vitality has decreased over the last decades [12,15,19]. In this study, detailed information on the piglet vitality was assessed. The umbilical cord is a good indicator for piglet vitality and intrapartum asphyxia [14,37,39]. The prevalence of ruptured umbilical cords or abnormalities varies widely from 20% to 70% in literature [30,38,41]. Abnormalities of the umbilical cord are considered as risk factor for stillbirth [30] and for an increased piglet mortality rate during lactation [41]. The results of our study on ruptured umbilical cords (<15%) in both treatment groups are lower than the described prevalence in several studies [30,38,41]. It can be hypothesized that this low prevalence of ruptured umbilical cords might be influenced by the free farrowing system [49]. However, a higher percentage of short ruptured umbilical cords in piglets of the 1,25-vitD group (vs. negative control group) was identified. Although the reason for these differences remains unclear, it can be hypothesized that 1,25-vitD may have affected the frequency, amplitude, or duration of uterus contractions in comparison to the control group by influencing calcium–phosphate homeostasis [5,46]. These parameters are known risk factors that influence umbilical cord traits, especially premature or short rupture of the umbilical cord [21,30,39]. Interestingly, in comparison with other studies [14,30,31,36,40,41], a short umbilical cord was not associated with a severe meconium staining or more stillborn piglets type 2 in the affected animals. Hence, it can be concluded that 1,25-vitD did not negatively affect the piglet vitality.

There was a significant difference in stillborn piglets type 1 (*p* ≤ 0.01), with more mummified piglets in the 1,25-vitD group. These findings are in contrast with two experimental studies, where, in general, a decreased number of stillborn piglets with larger vitamin D doses (1400 and 2000 IU) compared to the smaller doses of vitamin D (200 and 800 IU) were observed [32], but also a decrease in stillborn piglet type 1 in comparison to the control group was recorded [29]. The cause for an increased number of mummified piglets with additional vitamin D in the feeding ratio remains unclear. However, it is known that an overdose of vitamin D is fetotoxic in mammals [9], and can lead to abortion at different stages of pregnancy in sows [10]. Furthermore, the litter size is positively associated with the number of stillborn piglets type I [18,22,25] due to limitation of the uterine space. In addition, the blood supply to the uterine horns and to the placentae decreases in litters of more than 10 piglets and, therefore, the risk of an increased number of stillborn piglets type 1 is evident [50]. In summary, several reasons can lead to an increased number of stillborn piglets type 1 [17], and the detailed mechanism of 1,25-vitD remains unclear and warrants further studies.

## 5. Conclusions

This experimental field study showed the effect of a natural source of 1,25-vitD on the farrowing process with influence on piglet vitality. A trend in reducing the farrowing duration could be detected in the 1,25-vitD group. Furthermore, a significantly higher number of short ruptured umbilical cords and stillborn piglets type 1 were detected in the 1,25-vitD group. However, more research is needed to describe the mechanism of 1,25-vitD in detail and to evaluate the cause for an increased number of mummified piglets.

## Figures and Tables

**Figure 1 animals-12-00611-f001:**
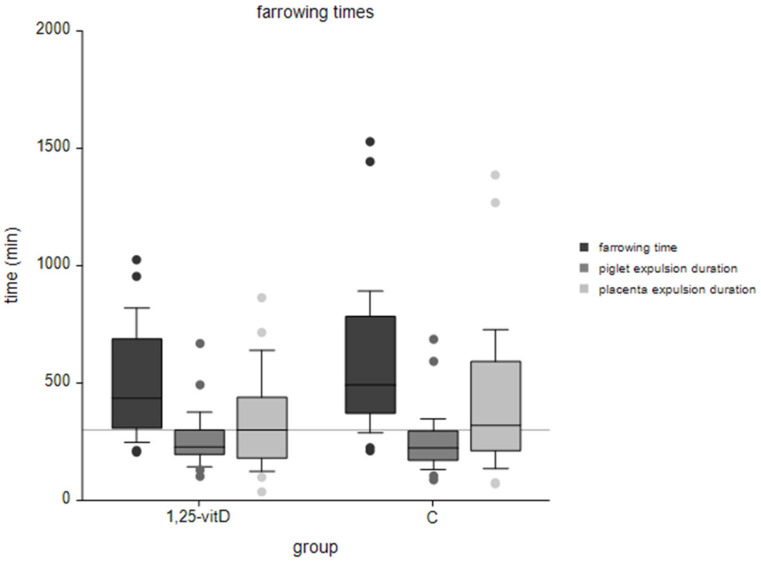
Assessed farrowing times grouped into 1,25-vitD and negative control group (C); gray horizontal line marks 300 min; horizontal line in boxplots were medians; ‘farrowing time’ means total farrowing time from the first piglet to the last placenta expelled, ‘piglet’ and ‘placenta expulsion duration’ means the time from the first to the last piglet/placenta expelled.

**Table 1 animals-12-00611-t001:** Fecal score, body condition score, and backfat thickness of the two study groups (negative Control group, received no complementary feed, *n* = 41 and 1,25-vitD group, received a defined amount of 1,25-dihydroxyvitamin D3, *n* = 48).

Parameters	Control (%)	1,25-vitD (%)	*p*-Value
Fecal score			
0	7.3	2.1	0.90
2	34.2	41.7	
3	56.1	56.2	
4	2.4	0.0	
Body condition score			0.39
2	4.9	2.1	
3	46.3	60.4	
4	48.8	37.5	
Backfat thickness *	14.8 ± 4.3	15.4 ± 4.0	0.51

Scores show the percentages, marked with an asterisk (*) are normally distributed parameters with mean ± standard derivation calculated.

**Table 2 animals-12-00611-t002:** Descriptive data of farrowing and sow traits of the two study groups (negative control group, received no complementary feed, *n* = 41 and 1,25-vitD group, received a defined amount of 1,25-dihydroxyvitamin D3, *n* = 48).

Farrowing Traits	Control	1,25-vitD	*p*-Value
Piglet interval (min) *	19.5 ± 10.4	17.9 ± 7.0	0.72
Total farrowing duration (min) *	586 ± 304	494 ± 220	0.17
Piglet expulsion duration (min) *	248 ± 114	253 ± 101	0.84
Placenta expulsion duration (min) *	407 ± 292	328 ± 186	0.38
First piglet–first placenta (min) *	179 ± 68	181 ± 75	0.99
Last piglet–last placenta (min) *	338 ± 281	253 ± 180	0.24
Last piglet–first placenta (min) *	−69 ± 106	−72 ± 76	0.86
Obstetrical intervention conducted	58.5%	43.8%	0.16

Mean ± standard derivations were calculated; marked with an asterisk (*) are normally distributed parameters.

**Table 3 animals-12-00611-t003:** Descriptive data of piglet traits of the two study groups (negative control group, received no complementary feed, *n* = 41 and 1,25-vitD group, received a defined amount of 1,25-dihydroxyvitamin D3, *n* = 48).

Piglet Traits	Control	1,25-vitD	*p*-Value
Total born piglets	16 (4; 23)	16 (5; 26)	0.30
Live born piglets	15 (4; 22)	15 (5; 24)	0.68
Stillborn piglets type 1	**0** (**0**, **1**) **	**0** (**0**, **3**) **	**0.01** **
Stillborn piglets type 2	0 (0, 2)	0 (0, 7)	0.47
Litter weight *	22.1 ± 5.0	21.1 ± 4.5	0.35
Individual pig weight *	1.6 ± 0.3	1.5 ± 0.3	0.16
Placenta weight *	4.4 ± 1.1	4.1 ± 1.2	0.39
Intact umbilical cord (%)	85.8	88.5	0.30
Ruptured umbilical cord (%)	14.2	11.5	
Long umbilical cord (%)	96.9	94.2	0.07
Short umbilical cord (%)	3.1	5.8	
Meconium staining score 1 (%)	64.4	70.4	0.28
Meconium staining score 2 (%)	29.8	25.3	0.33
Meconium staining score 3 (%)	5.8	4.3	0.34

Median (minimum; maximum) or mean ± standard derivations were calculated; marked with an asterisk (*) are normally distributed parameters, bold ** are significant values.

**Table 4 animals-12-00611-t004:** Multiple regression model with log(farrowing duration) as dependent variable.

Independent Variable	Regression Coefficient	*p*-Value	Power of Test at 5%	Regression Coefficient Invert Function (10^×^)
Intercept	2.498	**0.000** **	1.000	314.60
Total born piglets	0.015	**0.004** **	0.827	1.03
‘1,25-vitD’	−0.079	0.055	0.484	−1.20

Bold ** are significant values.

**Table 5 animals-12-00611-t005:** Mixed-effect logistic regression for piglet vitality parameters with ‘sow’ as random effect.

Piglet Vitality Parameter	Variable	Coefficient	*p*-Value	95% Confidence Interval
Umbilical cord length	1,25-vitD/C	0.695	**0.042** **	0.026 to 1.36
	Sow (random effect)	0.557		0.173 to 1.792
Umbilical cord integrity	1,25-vitD/C	−0.238	0.377	−0.767 to 0.291
	Sow (random effect)	0.840		0.451 to 1.563
Meconium score	1,25-vitD/C	−0.340	0.191	−0.850 to −0.348
	Sow (random effect)	1.074		0.674 to 1.711

Bold ** are significant values.

## Data Availability

Not applicable.

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
