# Peer review of "Effect of 1,25-Dihydroxyvitamin D3-Glycosides on the Farrowing Process and Piglet Vitality in a Free Farrowing System"

_animals, 2022, doi:10.3390/ani12050611_

Round 1

Reviewer 1 Report

In the present manuscript, the authors perform a study to evaluate the effects of 1,25-dihydroxyvitamin D3 (1,25vitD) on the farrowing rates and piglets traits under free farrowing system conditions. Sows in the problem group received a fixed dosage of 1,25vitD during gestation period while sows in the control group did not received any feed supplementation. Parameters related to the sow as backfat thickness, fecal score and farrowing process were recorded. Numerous parameters regarding piglet performance were also analyzed namely, total and live born piglets, born or stillborn, birth individual and total litter weight, and umbilical cord and meconium score as indicators of piglets viability. The results, despite a limited sample size, seem to indicate a positive effect of 1,25vitD on the farrowing process with a reduced  duration of the farrowing process and a higher number of total born piglet than those obtained in control group.  The methodological procedures applied seems adequate to achieve the proposed objective and the obtained results are interesting. However, and although, different aspects should be revised before the manuscript was ready for its publication.

COMMENTS TO THE AUTHORS

1.- Material and methods section: This section needs to be deeply revised and modified. In its current format is not adequate to be published since there is an important lack of information and the information included is presented in a not adequate way.

I suggest to the authors to divide this sections in subsections. In an example:

2.1.- Ethical issues:

2.2.- Animals and housing conditions

2.3.- Parameters evaluated

                        2.3.1.- Sows and farrowing process traits

                        2.3.2.- Piglets traits

2.4.- Experimental design

2.5.- Statistical analysis

Within each of these subsection, please include more detailed information namely, details free farrowing system, detailed description of meconium and umbilical cord evaluation scores, better explanation about evaluation of interval between piglet an placenta expulsion. In summary, all method used for recorded the data should be better explained.

In the same line, experimental design should be also improved. It seems that five batches of AI were performed. The time period when these inseminations were performed should be indicated. Additionally, why this dose of 1,25vitD was used?. Do you have information from previous experiments that support the use of the selected concentration?.

9.- Discussion section:  Despite the interesting results obtained, this section is, in my opinion, should be revised and improved to increase the quality of the manuscript. Mainly the discussion regarding stillborn piglets should be, in my opinion, revised. Is there any possible  relation between percentage of short umbilical cord and frequency of stillborn piglets?. I would be very grateful to the authors if they could go deeper in these aspects.

10.- Discussion; page 7, line 255: I would suggest to change “study” by “studied”. Please revise it and modify if convenient.

10.- Discussion; page 8, lines 271 to 274: Please revise these lines. Is this the correct meaning?. It seems that is something lacking in the sentence. Please, revise it and modify if convenient.

10.- Discussion; page 7, line 291: Is server the correct word here?. May be “severe” is the correct one. Please, revise it and modify if convenient.

10.- Conclusion: Please remove the p value from the conclusion

11.- Tables: Tables 2 and 3. Please include the title in the column of parameters evaluated as in Tables 4 and 5.   of FPKM. Although it is defined in the main text it should be also indicated here.

Reviewer 2 Report

Brief Summary: This original research paper describes how 1,25-dihydroxyvitamin D3 (1,25vitD) affected the farrowing process in sows and their piglets’ vitality. The paper’s novelty was the authors’ approach to evaluate supplementation effects of 1,25vitD on the farrowing process and farrowing time on piglets’ vitality in a free farrowing system. The paper’s strength is the detailed statistical analyses to explore the 1,25vitD supplementation effects. Although most of the reported findings were not statistically different to illustrate 1,25vitD’s supplementation effects, these important data hold strong impact for future investigations on vitamin D’s impact on farrowing and piglet vitality. The authors noted a limitation of the study was the sample size, but considering the scope of the objectives of this study, this limitation does not detract from the overall conclusions. This exciting paper has important impact for improving farrowing!

Major comments:

The statistical analyses section is detailed extensively well for readers to understand how data was analyzed in this study!

Was serum collected from the sow to measure serum vitamin D? What about measuring vitamin D from the umbilical cord?

Lines 44 – 46: It is not clear why the vitamin D receptor is mentioned in the Introduction section. Although its effect with vitamin D is described, the study did not report measuring vitamin D receptor expression and could potentially cause readers confusion who may think there will be data on vitamin D receptor. Can these sentences be moved to the Discussion and highlighted that vitamin D receptor can be measured for follow-up experiments relating to this study’s objectives?

Minor Comments:

Although the parity of the sows is mentioned in the beginning of the results, it would be helpful for readers if for those details can also be included in the Materials and Methods section where such details would come sooner where the experiment procedures is described.

What was the age of the sows used in this study? If the increase in litter size and prolonged farrowing cause reduced piglet vitality, then the physiological demands on the sow are likely driving such an effect.

Round 2

Reviewer 1 Report

The authors have addressed all the changes proposed and have answered all the questions raised. The manuscript have been improved notably and can be published in its current form.